# Non-oncological gynecological diagnoses in a women's health care service during the pandemic caused by the Severe Acute Respiratory Syndrome Coronavirus 2

**Laís Ribeiro Coca Parada[1], José Antonio Orellana Turri[1], Valery Helena da Costa[1,2], Ingrid Batista Vieira[1,3], Edmund Chada Baracat[1], José Maria Soares Júnior[1], Isabel Cristina Esposito Sorpreso[1]***

1 Departamento de Obstetricia e Ginecologia, Disciplina de Ginecologia, Faculdade de Medicina FMUSP, Universidade de São Paulo, São Paulo, SP, Brazil, 2 Faculdade Israelita de Ciências da Saúde Albert Einstein, São Paulo, SP, Brazil, 3 Faculdade de Saúde Pública da Universidade de São Paulo, Universidade de São Paulo, São Paulo, SP, Brazil

* icesorpreso@usp.br

**Data Availability Statement:** Harvard dataverse, https://doi.org/10.7910/DVN/8TRFZM.

## Abstract

### Objective

Analyze clinical factors and non-oncological gynecological diagnoses before and during the initial months of the COVID-19 pandemic.

### Method

Crosssectional study at an Outpatient Gynecology Clinic in Brazil involving medical consultations performed during the pre-pandemic and pandemic periods. The number of visits, prevalence of non-oncological gynecological diagnoses, and clinical-demographic data were analyzed. Parametric continuous variables were evaluated by Student's t-test and ANOVA tests, non-parametric variables were evaluated by the Mann-Whitney and Wilcoxon tests, and categorical or binary variables were evaluated by chi-square and Fisher's exact tests. Univariate logistic regression tests were performed, and variables with p ≤ 0.20 were subjected to multivariate logistic regression. Statistical significance was set at p < 0.05.

### Results

There were 1,236 records during the pre-pandemic period and 530 during the pandemic, reflecting a significant reduction (57.88%; p = 0.001) in medical consultations. The outpatient prevalence of women older than 50 y (OR 0.85; 95%CI 0.68–1.05) reduced, and the outpatient prevalence of postmenopausal women with hot flashes (OR 1.34; 95%CI 1.09–1.65; p = 0.005) and alcohol consumption habits (OR 2.76; 95%CI 1.15–6.59; p = 0.023) increased. There was a 6% proportional increase in noninflammatory disorders of the female genital tract (p = 0.030) and a 72.4% decrease in general physical examinations, contraception, and procreation (p = 0.001). Multivariate analysis showed that there was an

**Funding:** The author(s) received no specific funding for this work.

**Competing interests:** The authors have declared that no competing interests exist.

increased prevalence of abnormal uterine bleeding (OR, 1.7; 95% CI 1.34–2.16; p = 0.001) and endometriosis (OR 1.65; 95% CI 1.13–2.42; p = 0.01).

## Conclusion

Medical consultations for benign gynecological diseases during the pandemic prevented non-inflammatory disorders of the female genital tract, with an emphasis on abnormal uterine bleeding and endometriosis. There was an increased prevalence of women under 50 years of age, women with symptoms of hot flashes, and alcohol consumption habits and a reduction in the prevalence of general physical examinations, contraception, and procreation.

## Introduction

Severe Acute Respiratory Syndrome Coronavirus 2 (SARS-CoV-2) originated in Wuhan, China, in December 2019 and spread worldwide, causing the World Health Organization to declare a global pandemic [1]. Until march 2022, Brazil was the third most affected country, with more than 29 million confirmed cases and 655 thousand deaths [2], while, São Paulo was the most affected state in Brazil, with more than 5 million confirmed cases and 166 thousand deaths [3].

During the pandemic, the Brazilian Health Regulatory Agency recommended postponing non-emergency procedures and consultations [4]. Outpatient visits and returns, specifically for women's health, were recommended on a case-to-case assessment, considering the general and clinical status of women and the access to clinical treatment and resources [4]. Notwithstanding these recommendations, there was also a reduction in the spontaneous search for health services [5, 6], with a general decrease in procedures and diagnostic tests in the initial phase of the pandemic in Brazil and in other countries [7].

In gynecology, there is little information in the medical literature comparing the general demand for outpatient care before and during a pandemic [7]. In Brazil, the main gynecological diagnoses are menstrual disorders and abnormal bleeding from the female genital tract, inflammatory processes, and urogenital and breast disorders [8, 9] and understanding the changes caused by the pandemic on women's healthcare services is important to support the appropriate decision-making in the management of health care and resources [5, 10, 11].

The characterization of the assisted population in the different levels of health services contributes to the quality of health care, which results in topics pertinent to women's health in outpatient interdisciplinary training and fundamental points in the hierarchy of health services in which promotion and treatment measures are still incipient [11, 12].

In addition, characterizing the assisted population and general prevalence rates of non-oncological gynecological diagnoses and the clinical factors involved are important, since the Brazilian Unified Health System works in a regionalized and equitable manner [12]. Thus, the objective of this study was to analyze clinical factors and non-oncological gynecological diagnoses before and during the initial months of the COVID-19 pandemic.

## Materials and methods

### Study design and settings

This cross-sectional study was conducted at the Outpatient Gynecology Clinic of Hospital das Clínicas da Faculdade de Medicina da Universidade de São Paulo (HCFMUSP), São Paulo,

Brazil, between January 2019 and April 2021. This hospital corresponds to levels of tertiary and quaternary care that treat women of high complexity from several health establishments of the Unified Health System in Brazil, in addition to being dedicated to undergraduate teaching, medical and multiprofessional residency, and research. Corresponding to protocol 228/13, this study was approved by the Ethics Committee of our institution. The Ethics Committee renounced the need for informed consent.

## Participants

Secondary data were collected from 1,766 medical records in an electronic format, with 1,236 medical records of consultations conducted prior to the confinement decree of São Paulo (01/01/2019 to 03/22/2020), and 530 records of consultations made after the decree was established, the period referred to here as during the pandemic (03/22/2020 to 04/08/2021). This sample corresponds to all services provided by the health service during the given period.

The inclusion criteria were women over 18 years of age. The exclusion criteria were diagnoses of pregnancy, childbirth, puerperium, miscarriage, ectopic pregnancy, diagnosis related to trauma, and oncological diseases.

During the pandemic, HCFMUSP, an important hospital in Latin America, received critically ill patients with SARS-CoV-2, with 900 beds intended for their management. In this context, outpatient clinics have reduced their capacity, including gynecological clinics [13].

## Study size

The optimal sample size required for this study was calculated considering the magnitude of minimal difference of 10%, using the proportion between the total number of patients in the period before and during the pandemic, with an alpha level of 0.05, and a minimum estimation power of 80% [14, 15]. So, the sample sizes calculated for this study were 85 and 126, considering minimal differences of 10% and 1% in proportion, respectively, between the two periods of analysis.

## Data collection procedure

Data from electronic medical records were extracted from the hospital database.

Variables used in the clinical-sociodemographic data were age, body mass index, systolic blood pressure (SBP), diastolic blood pressure (DBP), age at menopause, parity, age at menarche, age at first intercourse, multimorbidity (having two or more chronic diseases), menopause (absence of menstrual cycles for longer than 12 months and hysterectomized patients > 40 y), sexual life (active or inactive), presence of hot flashes, smoking (yes or no), alcohol consumption habit (yes or no), and ethnicity (white and nonwhite).

Gynecological diagnoses were standardized according to the International Classification of Diseases in its tenth revision (ICD 10, 1997). Five disease categories were grouped: 1) Diseases of the urinary system (N30-N39) including urinary incontinence, cystitis, neuromuscular disorders of the bladder, other disorders of the urinary system, urethritis and urethral syndrome, urethral stricture, other urethral disorders, and bladder disorders; 2) Disorders of breast (N60-N64) including benign mammary dysplasias, inflammatory disorders of breast, hypertrophy of breast, unspecified lump in breast, fistula and fissure of the nipple, fatty necrosis of the breast, atrophy of breast, non-associated to birth galactorrhea, mastodynia, and solitary cyst of the breast; 3) Inflammatory diseases of female pelvic organs (N70-N77) including lower genital tract infections (herpes, gonorrhea, chlamydia, Trichomonas, Candida, vulvovaginitis, and syphilis), infectious vulvar lesions, inflammatory diseases of the upper genital tract, such as diseases of the uterus, ovaries, fallopian tubes including cervicitis,

salpingitis, endometritis, and tube-ovarian abscess, diseases of Bartholin gland, and vulvovaginal ulceration and inflammation; 4) Non-inflammatory disorders of female genital tract (N80-N99) including genital dysplasia (precancerous lesions of the vulva, vagina, and cervix), menopausal disorders, menstrual disorders and hormonal dysfunction (dysfunctional uterine bleeding, ovarian hyperestrogenism, ovarian dysfunction, or irregular menstrual cycles), endometriosis, malignancy of the reproductive tract (carcinoma in situ and invasive diseases of the genital tract), benign disorders of the uterus and ovaries (benign ovarian cysts or tumors, leiomyomas, endometrial polyps, and hyperplasia), and infertility; 5) General physical examination, contraception, and procreation (Z00-Z31) including general examination and investigation of people without complaints regarding contraception, general advice about contraception, insertion of contraceptive devices (intrauterine), sterilization, and measures of procreation. These categories are the same as those used in previous studies by our group [16, 17]. In cases where a single patient presented with two or more diagnoses, each was described separately.

## Bias

To address potential sources of bias, the authors considered that all data from electronic medical records were extracted from the hospital database. In cases where a single patient presented with two or more diagnoses, each was described separately. When information regarding a specific variable of the analysis was not described in its categorical or numeric field, the authors searched the complementary free fields for the missing values. Data were checked for consistency after data collection, and medical records were re-assessed (by LRCP and ICES) when there were discrepancies.

## Statistical analysis

Data were entered into Microsoft Excel 2016, São Paulo, Brazil and analyzed by three researchers (LRCP, JAOT, and ICES). All analyses were performed using the Stata program (Stata 16SE).

Clinical-demographic data were presented as means, standard deviations, medians, quartiles, and their frequency of involvement in percentage. The statistical analysis did not include all variables that present a greater number of missing data. When possible and clinically relevant, variables with a greater number of missing data were combined and included in the statistical analysis as multi-variables or grouped variables. Prior to analysis of differences between pre- and during pandemic periods, distribution tests were performed (Shapiro-Wilk and Smirnov-Kolmogorov), parametric continuous variables were evaluated by Student's t-test and ANOVA and non-parametric variables by the MannWhitney and Wilcoxon tests, and categorical or binary variables were analyzed using chisquare and Fisher's exact tests.

Regression analyses were performed to determine the main characteristics associated with the decrease in the number of medical consultations during the pandemic. Univariate logistic regression tests were performed, and variables with $p \leq 0.20$ were classified by the degree of significance and subjected to multivariate logistic regression using the stepwise forward methodology; variables with $p < 0.05$ were found to be significantly associated with the change (decrease or increase) in the number of medical consultations during the pandemic.

The optimal sample sizes calculated for this study were 85 and 126, considering minimal differences of 10% and 1% in proportion, respectively, between the two periods of analysis. Statistical significance was set at $p < 0.05$, and all tests and analyses were conducted using the STATA-16 SE software.

## Results

Of the initial 1,768 medical records collected, two were excluded (one pregnant patient and one patient that underwent abortion). There were 1,236 pre-pandemic and 530 pandemic records, which revealed a significant proportional reduction of medical consultation 57.88% (p = 0.001). Regarding the 1,236 patients included in this study, the power sample was 100%, considering statistically significant p-values of 5% and 1% in both cases.

Among the variables of clinical-demographic characteristics of patients, decrease in mean age (from 54.20 ± 10.15 to 53.01 ± 9.48), increase in mean SBP (from 120.71 ± 18, 27 to 126.23 ± 17.36) and DBP (72.2 ± 11.03 to 77.52 ± 10.61) measured in the outpatient ambulatory, increase in the number of women with hot flashes (35.44%, n = 438, to 42.45%, n = 225), alcohol consumption habits (from 0.81%, n = 10, to 2.08%, n = 11), and multimorbidity ($\geq$ 2 comorbidities) (from 50.89%, n = 629, to 57.17%, n = 303) revealed significant changes with p-values < 0.05 (Table 1).

The prevalence of diagnoses during the pre-pandemic patient visits was as follows: non-inflammatory disorders of the female genital tract (76.86%, n = 950), general physical examination, contraception, and procreation (17.88%, n = 221), disorders of the breast (17.31%, n = 214), diseases of the urinary system (7.77%, n = 96), and inflammatory diseases of the female pelvic organs (4.21%, n = 52). During the pandemic, the prevalence was as follows: non-inflammatory disorders of the female genital tract (81.51%, n = 432), disorders of the

**Table 1. Clinical-sociodemographic characteristics of women at the HCFMUSP gynecology outpatient clinic pre- and during pandemic.**

| Variable | | Pre-pandemic | | During pandemic | | P-value |
|---|---|---|---|---|---|---|
| | | Med ±Sd | Median (Q1 -Q3) | Med ±Sd | Median (Q1 -Q3) | |
| Age | | 54.20 ± 10.15 | 55 (49–60) | 53.01 ± 9.48 | 53 (49–58) | **0.0028** |
| Body Mass Index | | 29.23 ± 5.25 | 28.53 (25.40–32.79) | 29.32 ± 5.49 | 28.93 (25.03–32.76) | 0.9424 |
| Systolic Blood Pressure (mmHg) | | 120.71 ± 18.27 | 119 (108–130 | 126.23 ± 17.36 | 123 (114–139) | **0.0113** |
| Diastolic Blood Pressure (mmHg) | | 72.2 ± 11.03 | 71 (65–80) | 77.52 ± 10.61 | 80 (70–83) | **0.0007** |
| Age of Menopause | | 46.37 ± 7.07 | 48 (43–51) | 46.79 ± 7.00 | 48 (43–51) | 0.6389 |
| Parity | | 2.72 ± 1.50 | 3 (2–3) | 2.82 ± 1.34 | 3 (2–4) | 0.205 |
| Age of Menarche | | 13.15 ± 2.54 | 13 (12–14) | 13.14 ± 2.03 | 13 (12–14) | 0.5491 |
| Age at first intercourse | | 20.61 ± 7.82 | 17 (16–24) | 19.02 ± 5.49 | 17 (16–19) | 0.8455 |
| | | N | % | N | % | |
| Multimorbidity | Yes | 629 | 50.89 | 303 | 57.17 | **0.015** |
| | No | 607 | 49.11 | 227 | 42.83 | |
| Menopause | Yes | 586 | 47.41 | 266 | 50.19 | 0.284 |
| | No | 650 | 52.59 | 264 | 49.81 | |
| Active sexual life | Active | 338 | 60.57 | 173 | 64.79 | 0.243 |
| | Inactive | 220 | 39.43 | 94 | 35.21 | |
| Hot flashes | Yes | 438 | 35.44 | 225 | 42.45 | **0.005** |
| | No | 798 | 64.56 | 305 | 57.55 | |
| Smoking | Yes | 61 | 4.94 | 28 | 5.28 | 0.759 |
| | No | 1175 | 95.06 | 502 | 94.72 | |
| Drinking | Yes | 10 | 0.81 | 11 | 2.08 | **0.024** |
| | No | 1226 | 99.19 | 519 | 97.92 | |
| Ethnic group | White | 915 | 74.03 | 380 | 71.7 | 0.310 |
| | Non-white | 321 | 25.97 | 150 | 28.3 | |

HCFMUSP: Hospital das Clínicas da Faculdade de Medicina da Universidade de São Paulo; Sd: standard deviation; Med: median.

**Table 2. Prevalence of non-oncological gynecological diagnoses and the reduction of consultations pre- and during pandemic.**

| Number of consultations | Pre-pandemic | | During pandemic | | Reduction in the number of Consultations (%) | P-value |
|---|---|---|---|---|---|---|
| | N | % | N | % | | |
| Total | 1,236 | 100 | 530 | 100 | 57.1% | **0.001** |
| Diseases of the urinary system ICD (N30—N39) | 96 | 7.77 | 44 | 8.3 | 54.2% | 0.703 |
| Disorders of breast ICD (N60—N64) | 214 | 17.31 | 95 | 17.92 | 55.6% | 0.757 |
| Inflammatory diseases of female pelvic organs ICD (N70—N77) | 52 | 4.21 | 22 | 4.15 | 57.7% | 0.957 |
| Non-inflammatory disorders of female genital tract ICD (N80—N99) | 950 | 76.86 | 432 | 81.51 | 54.5% | **0.030** |
| General physical examination, contraception, and procreation ICD (Z00—Z31) | 221 | 17.88 | 61 | 11.51 | 72.4% | **0.001** |

ICD: International Classification Disease 10[th] edition.

breast (17.92%, n = 95), general physical examination, contraception, and procreation (11.51%, n = 61), diseases of the urinary system (8.30%, n = 44), and inflammatory diseases of female pelvic organs (4.15%, n = 22). The proportional increase in the prevalence of non-inflammatory disorders of the female genital tract was significant (p = 0.030), as was a decrease in the prevalence of general physical examination, contraception, and procreation (p = 0.001) (Table 2).

The number of consultations generally decreased in month 15, which corresponds to March 2020. A substantial decrease in the prevalence of general physical examinations, contraception, and procreation without substantial return corresponds to the darker blue portion shown in Fig 1.

SARS-CoV-2 (Severe Acute Respiratory Syndrome Coronavirus 2); HCFMUSP.

Among the non-inflammatory disorders of the female genital tract, abnormal uterine bleeding (AUB) and endometriosis were analyzed separately because of their individual prevalence. The prevalence of AUB was 18.28% (n = 226) in the pre-pandemic period and 27.90% (n = 148) during the pandemic period (p<0.001). Endometriosis had a prevalence of 5.66% (n = 70) before the pandemic period and 9.43% (n = 50) during the pandemic period (p = 0.004).

Categorical variables were subjected to a univariate logistic regression analysis. In this univariate logistic regression analysis, AUB (OR 1.73, 95% CI 1.36–2.20, p<0.001), general physical examination, contraception, and procreation (OR 0.60, 95% CI 0.44–0.81, p = 0.001), endometriosis (OR 1.74; 95% CI 1.19–2.53, p = 0.004), hot flashes (OR 1.34; 95% CI 1.09–1.65; p = 0.005), alcohol consumption habits (OR 2.6; 95% CI, 1.10–6.16; p = 0.030), non-inflammatory disorders of the female genital tract (OR 1.32; 95%CI 1.03–1.71; p = 0.030), and age > 50 y (OR 0.85; 95% CI 0.68–1.05; p = 0.133) were variables that, in relation to p-values, had p-values lower than 0.2 and followed to multivariate logistic regression model. Variables that did not follow were ethnicity (OR 0.89, 95% CI 0.71–1.12, p = 0.310), diseases of the urinary system (OR 1.08, 95% CI 0.74–1.56; p = 0.703), disorders of the breast (OR 1.04; 95%CI 0.80–1.36, p = 0.757), smoking (OR 1.07, 95% CI 0.68–1.70, p = 0.760), multimorbidity (OR 0.96, 95% CI 0.53–1.73, p = 0.880), and inflammatory diseases of the female pelvic organs (OR 0.99, 95% CI 0.59–1.64, p = 0.957).

Following the use of multivariate logistic regression, the variables that remained significant (p < 0.05) were AUB, endometriosis, alcohol consumption habits, and noninflammatory disorders of the female genital tract (Table 3). AUB showed a 70% increase in consultations, general physical examination, contraception, and procreation (38%), endometriosis (65%), and alcohol consumption (176%) (Table 3).

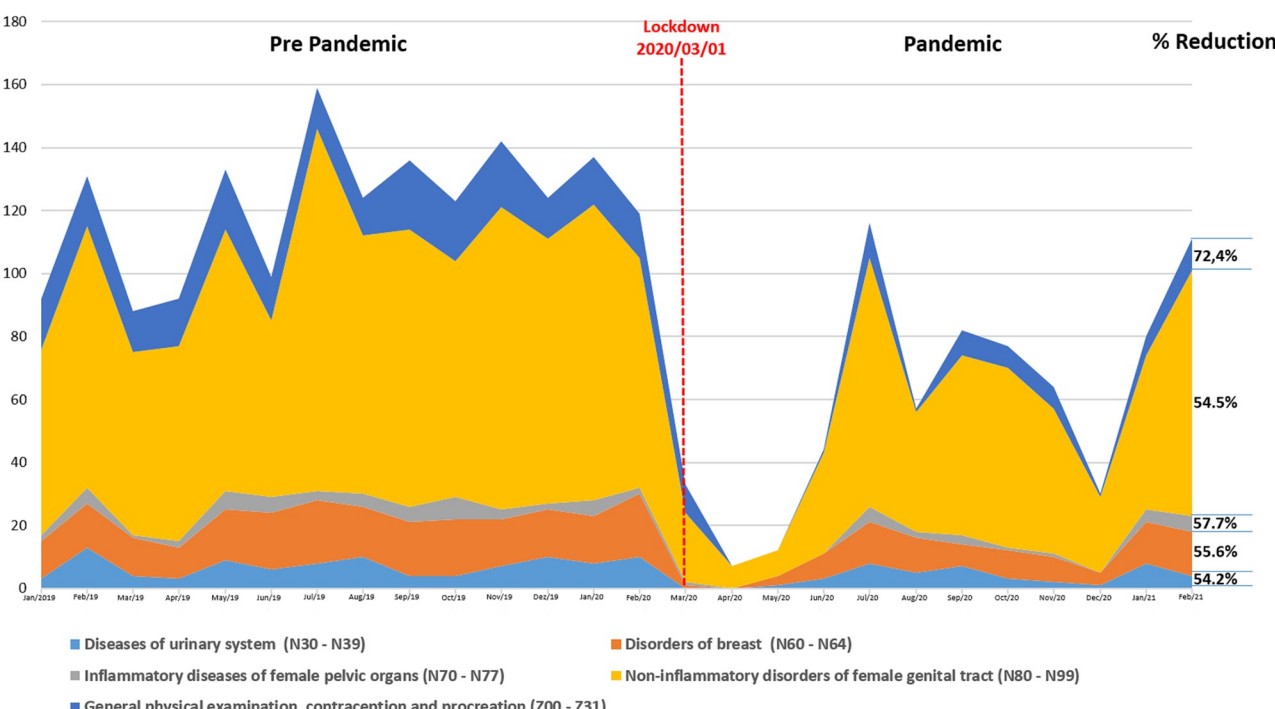

**Fig 1. Non-oncologic diagnosis number in pre- and SARS-CoV-2 pandemic period.** Proportional reduction in appointment in the HCFMUSP gynecologic outpatient care (Jan 2019 –Feb 2021). The widths of each color correspond to the total consultations of each diagnoses group per month.

## Discussion

The pandemic caused by SARS-Cov-2 brought about changes in women's health gynecology practice that involved postponed clinical consultations, elective surgical procedures, treatments, and a reduction in outpatient and non-essential health procedures. Our results indicated a change in the profile of women that were treated a specialized reference center for women's health treatment, with a focus on the gynecology practice of outpatient clinic that treats complex cases. Compared to the pre-pandemic period, women were younger, had greater infliction of chronic diseases, had a higher prevalence of alcohol consumption, and reported a greater perception of vasomotor symptoms, particularly hot flashes.

Patients with the aforementioned profile were less likely to discontinue their consultations during the pandemic, and the discontinuation is expected considering that old age is a risk factor for contracting a severe form of COVID-19 infection, and elderly patients are likely to avoid going to the hospital. Other possibilities of treatment discontinuation include

**Table 3. Multivariate logistic regression.**

| Variable | OR | CI 95% | P-value |
|---|---|---|---|
| Abnormal Uterine Bleeding | 1.70 | 1.34–2.16 | <**0.001** |
| General physical examination, contraception, and procreation | 0.62 | 0.46–0.85 | **0.003** |
| Endometriosis | 1.65 | 1.13–2.42 | **0.010** |
| Alcohol consumption habits | 2.76 | 1.15–6.59 | **0.023** |

the referral of patients to the tertiary service during the pandemic, or even an increase in the prevalence of these symptoms in the general population. In addition, the increase in the prevalence of alcohol consumption is in accordance with the findings from certain populations during the pandemic [18, 19]. Moreover, our findings on mean age at menarche and menopause are in line with other Brazilian studies evaluating these clinical demographic variables [9, 17].

Our results demonstrate an overall reduction of 57.1% in the number of consultations, possibly owing to population behavior and security measures imposed by health authorities during the pandemic [4], which has also been shown by studies in other medical fields [20]. General physical examination, contraception, and procreation were most affected, demonstrating gaps in care, particularly for women in the reproductive period, which may have future repercussions on sexual and reproductive health. A study in the United Kingdom showed lower access to contraception and greater number of unplanned pregnancies during the pandemic [21]. Regarding the general physical examination and screening, a report during the pandemic revealed a lower admission rate and lower rate of diagnosis of cervical dysplasia compared to that in the pre-pandemic period, during which fewer general examinations were performed [22].

Non-inflammatory diseases of the female genital tract were the most referenced non-oncological gynecological diagnoses for the analyzed service and, together with AUB and endometriosis, showed a significant increase in proportion during the pandemic. Moreover, both diseases are benign but have an impact on the quality of life of women. The reported prevalence for the first condition is high, ranging from 14–25% [23]; it was found to be 18.28% before the pandemic and 27.90% during the pandemic in this study. Endometriosis is a non-inflammatory disease of the lower genital tract that causes chronic pelvic pain, infertility, and chronic stress, mainly affecting women during the reproductive period. Concerning endometriosis, a study from Turkey showed that the majority of patients were afraid of endometriosis-related problems during the pandemic, and about half considered the quality of care for their condition to have been affected by the pandemic [24]. Furthermore, these patients were more likely to require treatment in complex outpatient clinics.

The limitations of this study include the impossibility of accessing records of care that took place outside the evaluated hospital complex, which does not include data on lesser complex cases or emergency care. However, an analysis of all available data was performed without significant losses or consolidated electronic records.

This study is novel as it assessed the changes in gynecological clinical practice during the SARS-CoV-2 pandemic. This may help devise more precise policies to deal with the gaps in care caused by the pandemic, optimizing resource allocation, and the possible consequences of decreased appointments related to general physical examinations, contraception, and procreation, including screening. This information also aids in promoting women's health and increasing preparedness for similar situations in the future [25].

## Conclusion

There was a reduction in the number of visits for non-oncological gynecological diagnoses during the pandemic caused by SARS-CoV-2. The sociodemographic and clinical profiles of consultations were the prevalence of women in perimenopause, presenting multimorbidity, and an increase in alcohol consumption. Regarding nononcological gynecological diagnoses, there was a higher prevalence of non-inflammatory disorders of the female genital tract increasing in proportion during the pandemic, with a significant proportional increase in the care of women with a diagnoses of AUB and endometriosis. Furthermore, there was a

significant reduction in gynecological general physical examination, contraception, and procreation, which highlights the gaps in screening and reproductive planning.

## Author Contributions

**Conceptualization:** Laís Ribeiro Coca Parada, José Antonio Orellana Turri, Ingrid Batista Vieira, Edmund Chada Baracat, José Maria Soares Júnior, Isabel Cristina Esposito Sorpreso.

**Data curation:** Laís Ribeiro Coca Parada, José Antonio Orellana Turri, Isabel Cristina Esposito Sorpreso.

**Formal analysis:** José Antonio Orellana Turri, Isabel Cristina Esposito Sorpreso.

**Funding acquisition:** Isabel Cristina Esposito Sorpreso.

**Investigation:** Laís Ribeiro Coca Parada, José Antonio Orellana Turri, Valery Helena da Costa, Ingrid Batista Vieira, Edmund Chada Baracat, José Maria Soares Júnior, Isabel Cristina Esposito Sorpreso.

**Methodology:** Laís Ribeiro Coca Parada, José Antonio Orellana Turri, Ingrid Batista Vieira, Edmund Chada Baracat, José Maria Soares Júnior, Isabel Cristina Esposito Sorpreso.

**Project administration:** Edmund Chada Baracat, José Maria Soares Júnior, Isabel Cristina Esposito Sorpreso.

**Resources:** Edmund Chada Baracat, José Maria Soares Júnior.

**Software:** Edmund Chada Baracat, José Maria Soares Júnior, Isabel Cristina Esposito Sorpreso.

**Supervision:** Edmund Chada Baracat, José Maria Soares Júnior, Isabel Cristina Esposito Sorpreso.

**Validation:** Edmund Chada Baracat, José Maria Soares Júnior, Isabel Cristina Esposito Sorpreso.

**Visualization:** Edmund Chada Baracat, José Maria Soares Júnior, Isabel Cristina Esposito Sorpreso.

**Writing – original draft:** Laís Ribeiro Coca Parada, José Antonio Orellana Turri, Valery Helena da Costa, Ingrid Batista Vieira, Isabel Cristina Esposito Sorpreso.

**Writing – review & editing:** Edmund Chada Baracat, José Maria Soares Júnior, Isabel Cristina Esposito Sorpreso.

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
