## [Decision Letter · Decision Letter 0]

14 Nov 2022

PONE-D-22-10074Non-oncological gynecological diagnoses in a women’s health care service during the pandemic caused by the Severe Acute Respiratory Syndrome Coronavirus 2 (SARS- CoV-2).PLOS ONE

Dear Dr. Sorpreso,

Thank you for submitting your manuscript to PLOS ONE. After careful consideration, we feel that it has merit but does not fully meet PLOS ONE’s publication criteria as it currently stands. Therefore, we invite you to submit a revised version of the manuscript that addresses the points raised during the review process.

We look forward to receiving your revised manuscript.

Kind regards,

Juan A López-Rodríguez

Academic Editor

PLOS ONE

https://journals.plos.org/plosone/s/fileid=ba62/PLOSOne_formatting_sample_title_authors_affiliations.pdf.

Reviewers' comments:

Reviewer's Responses to Questions

**Comments to the Author**

1. Is the manuscript technically sound, and do the data support the conclusions?

Reviewer #1: Yes

2. Has the statistical analysis been performed appropriately and rigorously? 

Reviewer #1: Yes

3. Have the authors made all data underlying the findings in their manuscript fully available?

Reviewer #1: Yes

4. Is the manuscript presented in an intelligible fashion and written in standard English?

Reviewer #1: Yes

5. Review Comments to the Author

Reviewer #1: I would like to thank you for the opportunity to review this manuscript. The study “Non-oncological gynecological diagnoses in a women’s health care service during the pandemic caused by the Severe Acute Respiratory Syndrome Coronavirus 2 (SARSCoV-2)” addresses an interesting and new topic aiming at analyze the sociodemographic and clinical factors and non-oncological gynecological diagnoses before and during the initial months of the COVID-19 pandemic in women that were assessed in a tertiary reference outpatient clinic. However, there are some issues that before publication should be solved. Please, find below my comments:

Comments

1. The abstract is good, it brings the main information of the manuscript.

2. The introduction is well written, but I cannot identify a clear hypothesis.

3. The objective is correctly worded: clear and succinct and with specialized terms suitable for a general journal

4. The materials and methods described are appropriate to achieve the proposed objectives; are described clearly, completely, and succinctly to answer the proposed question. As for the data collection process and the instruments used, they are clearly described, but according to STROBE Statement we should report:

- Inclusion criteria should be more detailed (women over 18 years of age?)

- Bias: Describe any efforts to address potential sources of bias

- Study size: Information on study size is separate. The calculation result is in the first sentence of the results section, it should be in the method.

5. Statistical analysis is adequate but according to STROBE Statement we should report “Explain how missing data were addressed”

6. Results are clear, appropriate, and correctly presented. The tables contain useful information and are arranged accordingly, but:

- Table 1 appears to be missing information. Just below "Age at first intercourse" there are several "yes" and "no", but it's not clear where they belong.

- Figure 1 is low resolution

7. It presents and discusses the points of convergence and divergence in relation to other authors. It mentions possible generalizations and/or practical applications or limitations from the results obtained.

Criteria for Publication – Plos One

1. The study presents the results of original research: Yes

2. Results reported have not been published elsewhere: Ok

3. Experiments, statistics, and other analyses are performed to a high technical standard and are described in sufficient detail: Yes

4. Conclusions are presented in an appropriate fashion and are supported by the data: Yes

5. The article is presented in an intelligible fashion and is written in standard English: I am not qualified to rate this question

6. The research meets all applicable standards for the ethics of experimentation and research integrity: Yes

7. The article adheres to appropriate reporting guidelines and community standards for data availability: There is no information on this issue

6. PLOS authors have the option to publish the peer review history of their article (what does this mean?). If published, this will include your full peer review and any attached files.

Reviewer #1: No

---

## [Author Response · Author response to Decision Letter 0]

31 Jan 2023

Dear Editor

PONE-D-22-10074

Non-oncological gynecological diagnoses in a women’s health care service during the pandemic caused by the Severe Acute Respiratory Syndrome Coronavirus 2 (SARS- CoV-2).

Fowling below answer to Reviewer #1 and Comments to the Author:

Reviewer #1: I would like to thank you for the review of this manuscript. Please, find below our answer:

Comments

1. The abstract is good, it brings the main information of the manuscript. 

Thank you.

2. The introduction is well written, but I cannot identify a clear hypothesis.

The authors rewrite and included:

“In gynecology, there is little information in the medical literature comparing the general demand for outpatient care before and during a pandemic [7]. In Brazil, the main gynecological diagnoses are menstrual disorders and abnormal bleeding from the female genital tract, inflammatory processes, and urogenital and breast disorders [8,9] and understanding the changes caused by the pandemic on women's healthcare services is important to support the appropriate decision-making in the management of health care and resources [5,10,11]. 

The characterization of the assisted population in the different levels of health services contributes to the quality of health care, which results in topics pertinent to women’s health in outpatient interdisciplinary training and fundamental points in the hierarchy of health services in which promotion and treatment measures are still incipient [11,12].

In addition, characterizing the assisted population and general prevalence rates of non-oncological gynecological diagnoses and the clinical factors involved are important, since the Brazilian Unified Health System works in a regionalized and equitable manner [12]. Thus, the objective of this study…..”

3. The objective is correctly worded: clear and succinct and with specialized terms suitable for a general journal

We rewrite and modified: 

“analyze clinical factors and non-oncological gynecological diagnoses before and during the initial months of the COVID-19 pandemic.”

4. The materials and methods described are appropriate to achieve the proposed objectives; are described clearly, completely, and succinctly to answer the proposed question. As for the data collection process and the instruments used, they are clearly described, but according to STROBE Statement we should report:

- Inclusion criteria should be more detailed (women over 18 years of age?)

“The inclusion criteria were women over 18 years of age. The exclusion criteria were diagnoses of pregnancy, childbirth, puerperium, miscarriage, ectopic pregnancy, diagnosis related to trauma, and oncological diseases.”

- Bias: Describe any efforts to address potential sources of bias.

 “Bias 

To address potential sources of bias, the authors included that all data from electronic medical records were extracted from the hospital database. In cases where a single patient presented with two or more diagnoses, each was described separately. 

When some information regarding a specific variable was not present in the hospital electronic database, the authors searched the complementary free fields for the missing values. Data were checked for consistency after data collection, and medical records were re-assessed (by LRCP and ICES) when there were discrepancies.”

- Study size: Information on study size is separate. The calculation result is in the first sentence of the results section, it should be in the method.

We rewrite and modified: 

“Study Size

The optimal sample size required for this study was calculated considering the magnitude of minimal difference of 10%, using the proportion between the total number of patients in the period before and during the pandemic, with an alpha level of 0.05, and a minimum estimation power of 80% [13,14]. So, the sample sizes calculated for this study were 85 and 126, considering minimal differences of 10% and 1% in proportion, respectively, between the two periods of analysis.”

5. Statistical analysis is adequate but according to STROBE Statement we should report “Explain how missing data were addressed”

We included:

“The statistical analysis did not include all variables that present a greater number of missing data. When possible and clinically relevant, variables with a greater number of missing data were combined and included in the statistical analysis as multi-variables or grouped variables.”

6. Results are clear, appropriate, and correctly presented. The tables contain useful information and are arranged accordingly, but:

- Table 1 appears to be missing information. Just below "Age at first intercourse" there are several "yes" and "no", but it's not clear where they belong.

Thank you. We included the missing information and add all the variables.

- Figure 1 is low resolution

We change the figure 1 to a higher resolution. 

All authors agree to the answer and grateful for the comments.

Best regards,

Isabel.

---

## [Decision Letter · Decision Letter 1]

7 Feb 2023

Non-oncological gynecological diagnoses in a women’s health care service during the pandemic caused by the Severe Acute Respiratory Syndrome Coronavirus 2 (SARS- CoV-2).

PONE-D-22-10074R1

Dear Dr. Sorpreso,

We’re pleased to inform you that your manuscript has been judged scientifically suitable for publication and will be formally accepted for publication once it meets all outstanding technical requirements.

Kind regards,

Juan A López-Rodríguez

Academic Editor

PLOS ONE

Additional Editor Comments (optional):

Reviewers' comments:

Reviewer's Responses to Questions

**Comments to the Author**

1. If the authors have adequately addressed your comments raised in a previous round of review and you feel that this manuscript is now acceptable for publication, you may indicate that here to bypass the “Comments to the Author” section, enter your conflict of interest statement in the “Confidential to Editor” section, and submit your "Accept" recommendation.

Reviewer #1: All comments have been addressed

2. Is the manuscript technically sound, and do the data support the conclusions?

Reviewer #1: Yes

3. Has the statistical analysis been performed appropriately and rigorously? 

Reviewer #1: Yes

4. Have the authors made all data underlying the findings in their manuscript fully available?

Reviewer #1: Yes

5. Is the manuscript presented in an intelligible fashion and written in standard English?

Reviewer #1: Yes

6. Review Comments to the Author

Reviewer #1: The authors answered all questions asked in the review. I consider a manuscript prepared for publication.

7. PLOS authors have the option to publish the peer review history of their article (what does this mean?). If published, this will include your full peer review and any attached files.

Reviewer #1: No

---

## [Editor Report · Acceptance letter]

14 Mar 2023

PONE-D-22-10074R1 

Non-oncological gynecological diagnoses in a women’s health care service during the pandemic caused by the Severe Acute Respiratory Syndrome Coronavirus 2. 

Dear Dr. Sorpreso:

I'm pleased to inform you that your manuscript has been deemed suitable for publication in PLOS ONE. Congratulations! Your manuscript is now with our production department. 

Kind regards, 

on behalf of

Dr. Juan A López-Rodríguez 

Academic Editor

PLOS ONE